# Fitness Estimation for Viral Variants in the Context of Cellular Coinfection

**DOI:** 10.3390/v13071216

**Published:** 2021-06-23

**Authors:** Huisheng Zhu, Brent E. Allman, Katia Koelle

**Affiliations:** 1Department of Biology, Emory University, Atlanta, GA 30322, USA; julie.zhu@emory.edu; 2Graduate Program in Population Biology, Ecology, and Evolution, Emory University, Atlanta, GA 30322, USA; brent.elliott.allman@emory.edu; 3Emory-UGA Center of Excellence for Influenza Research and Surveillance (CEIRS), Atlanta, GA 30322, USA

**Keywords:** within-host dynamics, viral evolution, influenza H5N1, viral modeling

## Abstract

Animal models are frequently used to characterize the within-host dynamics of emerging zoonotic viruses. More recent studies have also deep-sequenced longitudinal viral samples originating from experimental challenges to gain a better understanding of how these viruses may evolve in vivo and between transmission events. These studies have often identified nucleotide variants that can replicate more efficiently within hosts and also transmit more effectively between hosts. Quantifying the degree to which a mutation impacts viral fitness within a host can improve identification of variants that are of particular epidemiological concern and our ability to anticipate viral adaptation at the population level. While methods have been developed to quantify the fitness effects of mutations using observed changes in allele frequencies over the course of a host’s infection, none of the existing methods account for the possibility of cellular coinfection. Here, we develop mathematical models to project variant allele frequency changes in the context of cellular coinfection and, further, integrate these models with statistical inference approaches to demonstrate how variant fitness can be estimated alongside cellular multiplicity of infection. We apply our approaches to empirical longitudinally sampled H5N1 sequence data from ferrets. Our results indicate that previous studies may have significantly underestimated the within-host fitness advantage of viral variants. These findings underscore the importance of considering the process of cellular coinfection when studying within-host viral evolutionary dynamics.

## 1. Introduction

Zoonotic pathogens are often poorly adapted to their spillover hosts. Viral adaptation, however, can occur during epidemiological spread following spillover, resulting in increases in viral transmission potential as the pathogen establishes itself in the host population [1]. This was observed most notably in influenza viruses that have successfully established in humans (e.g., [2,3]). The pandemic coronavirus SARS-CoV-2 provides a more recent example, with variant lineages that are better adapted to human hosts (such as D614G [4]) emerging and replacing earlier viral lineages. Viral adaptations that improve transmission potential often arise from their effect on within-host replication dynamics. For example, mutations that enable viruses to replicate more efficiently within hosts (in particular, in transmission-relevant tissues) could enhance transmission potential, as could mutations that allow for a more effective evasion of the host immune response.

In vivo studies could in principle be used to identify mutations that improve viral fitness in a spillover host. For example, experiments using the ferret animal model identified a set of influenza A subtype H5N1 mutations that increase viral replication within the nasal turbinate of hosts (a transmission-relevant tissue) and also increase transmissibility [5,6]. The fitness effects of mutations such as these have been estimated by interfacing quantitative models with data on how variants carrying these mutations change in frequency over the course of infection [7,8,9]. However, these approaches assume that fitness is an individual-level property of a variant. While this may be the case when cells are only singly infected, many viral infections involve significant levels of cellular coinfection. For example, due to incomplete viral genomes, influenza viruses heavily rely on complementation to produce viral progeny [10,11,12]. High levels of cellular coinfection in other viruses, such as HIV, is also likely, given the pervasiveness of recombinant genomes that are identified during viral sequencing [13,14].

Cellular coinfection can impede the ability of high-fitness variants to rise to high frequencies within an infected host. This is because of the phenomenon of ‘phenotypic hiding’ [15,16]. Phenotypic hiding comes about as a consequence of viral protein products being shared within coinfected cells. Delivery of a viral genome carrying a highly beneficial mutation results in the production of a viral protein that can provide a replicative benefit to all of the viral genomes present in the coinfected cell. Similarly, a viral genome carrying a deleterious (and potentially even lethal) mutation can be rescued by protein products derived from coinfecting viral genomes. Cellular coinfection thus results in natural selection no longer acting on individual viral genomes, but instead on viral collectives. This effectively reduces the strength of selection, such that deleterious mutations are purged more slowly [17] and beneficial mutations are also fixed more slowly [18]. As a result, the extent of cellular coinfection impacts the dynamics of allele frequency changes in an infection and affects fitness inference.

Here, we first develop a set of mathematical models to project changes in the allele frequencies of viral variants within infected hosts. Our models specifically allow for cellular coinfection and the effect of phenotypic hiding on allele frequency changes. Using Bayesian inference approaches, we then demonstrate how these mathematical models can be interfaced with longitudinally sampled allele frequency data to jointly estimate the relative fitness of a variant and cellular multiplicity of infection levels. Finally, we apply our developed methods to estimate the fitness effect of an adaptive mutation that was identified in an influenza H5N1 experimental challenge study performed using the ferret animal model. Our findings indicate that the fitness effect of this mutation is considerably higher than previously estimated and that cellular coinfection precipitously slowed down the rate of within-host influenza virus adaptation.

## 2. Materials and Methods

### 2.1. Deterministic Within-Host Evolution Model

Several studies have used longitudinal allele frequency data to estimate the relative fitness of a mutant allele over a wild-type allele within an infected host or from passage studies [9,19,20]. None of these models, however, account for the impact that cellular coinfection can have on variant allele frequency changes over time. To accommodate cellular coinfection, we first start with an evolutionary model that projects allele frequencies from one viral generation to the next in the absence of coinfection:
(1)$$q^{m}\left(t_{g+1}\right)=\frac{q^{m}\left(t_{g}\right)e^{\sigma_{m}}}{q^{m}\left(t_{g}\right)e^{\sigma_{m}}+\left(1-q^{m}\left(t_{g}\right)\right)e^{\sigma_{w}}}$$
where $q^{m}\left(t_{g}\right)$ is the frequency of the variant (mutant) allele in viral generation *g*, $\sigma_{m}$ (with range −∞ to ∞) is the selective advantage/disadvantage of the focal mutation, and $e^{\sigma_{m}}$ (with range ≥0) is the relative fitness of the variant allele over the wild-type allele. The fitness of the wild-type allele ($e^{\sigma_{w}}$) is defined as 1. This model is a simplification of a model first presented in [9]. That model considers an arbitrary number of viral haplotypes and further incorporates de novo mutation in its projection of allele frequencies. Here, we ignore de novo mutation over the course of infection and limit our analysis to two viral haplotypes: a wild-type viral genotype and a variant genotype carrying a mutant allele at a single locus. We adopt these simplifications to focus attention on the effect of cellular coinfection in within-host evolution.

To extend this initial model to allow for the effect of cellular coinfection, we first assume that viral genomes enter cells independently of other viral genomes. Under this assumption, viral genomes are distributed across cells according to a Poisson distribution. Given a mean overall cellular multiplicity of infection (MOI) of *M*, the variant’s mean MOI in viral generation $t_{g}$ is simply given by $M_{m}=q^{m}\left(t_{g}\right)M$ and the wild-type virus’s mean MOI is simply given by $M_{w}={(1-q}^{m}\left(t_{g}\right))M$. The probability that a cell is infected with *k* variant viral genomes and *l* wild-type viral genomes is then:
(2)$$P\left(k,l\right)=\left(\frac{e^{-M_{m}}{\left(M_{m}\right)}^{k}}{k!}\right)\left(\frac{e^{-M_{w}}{\left(M_{w}\right)}^{l}}{l!}\right)$$


Under the assumption that viral protein products within cells have additive effects, the fitness of a viral genome present in a cell carrying *k* variant viral genomes and *l* wild-type viral genomes is given by:
(3)$$F\left(k,l\right)=\frac{k}{k+l}e^{\sigma_{m}}+\frac{l}{k+l}e^{\sigma_{w}}$$


Note that this fitness does not depend on whether the focal genome is a variant viral genome or a wild-type viral genome, since all viral genomes within a cell share their protein products and thus have the same fitness.

The realized mean fitness of a viral variant in the context of cellular coinfection is calculated by taking a fitness average of the viral variant across its cellular contexts:
(4)$$\bar{e^{\sigma_{m}}}=\frac{\sum_{k=0}^{\infty }\sum_{l=0}^{\infty }kP\left(k,l\right)F\left(k,l\right)}{M_{m}}$$


Similarly, and the realized mean fitness of the wild-type virus in the context of cellular coinfection is given by:
(5)$$\bar{e^{\sigma_{w}}}=\frac{\sum_{k=0}^{\infty }\sum_{l=0}^{\infty }lP\left(k,l\right)F\left(k,l\right)}{M_{w}}$$


Examination of these equations indicates that the realized mean fitness of the viral variant and of the wild-type virus approach $e^{\sigma_{m}}$ and $e^{\sigma_{w}}$, respectively, as cellular MOI becomes small, as expected. As cellular MOI becomes large, $\bar{e^{\sigma_{m}}}$ and $\bar{e^{\sigma_{w}}}$ converge in their values, as expected.

Variant allele frequency changes in the context of cellular coinfection can then be projected using a modified version of Equation (Equation 1), where realized mean fitnesses replace individual-level viral fitnesses:
(6)$$q^{m}\left(t_{g+1}\right)=\frac{q^{m}\left(t_{g}\right)\bar{e^{\sigma_{m}}}}{q^{m}\left(t_{g}\right)\bar{e^{\sigma_{m}}}+\left(1-q^{m}\left(t_{g}\right)\right)\bar{e^{\sigma_{w}}}}$$


### 2.2. Stochastic Within-Host Evolution Model

A recent study highlighted the important role that stochastic processes can play in shaping patterns of within-host viral evolution [21]. The extent to which in vivo viral evolution is governed by stochastic processes can be quantified by the within-host effective viral population size, which we here refer to as $N_{e}$. Low values of $N_{e}$ indicate that genetic drift plays a large role in shaping within-host viral populations, while high values of $N_{e}$ indicate that genetic drift plays a more minor role. The within-host effective viral population size $N_{e}$ for seasonal influenza A viruses has recently been estimated as being on the order of 30–70 [22]. Given estimates such as these, we develop here a stochastic version of the within-host evolutionary model presented in the previous section. For simplicity, in the equations below, we use *N* rather than $N_{e}$ to denote the effective viral population size. With a variant allele frequency of $q^{m}\left(t_{g}\right)$ in generation $t_{g}$, the variant’s effective population size is given by:
(7)$$N_{m}\left(t_{g}\right)=Nq^{m}\left(t_{g}\right)$$
and the effective population size of the wild-type virus in generation $t_{g}$ is given by:
(8)$$N_{w}\left(t_{g}\right)=N\left(1-q^{m}\left(t_{g}\right)\right)$$


Defining the number of target cells as *C*, the mean cellular multiplicity of infection is given by *N*/*C*, the mean cellular MOI of variant virus is given by $N_{m}/C$, and the mean cellular MOI of the wild-type virus is given by $N_{w}/C$. Under the same assumption as before that viral genomes enter cells independently of one another, we stochastically determine the distribution of variant viruses across target cells using a multinomial distribution with the event probability of being in a cell given by 1/*C* (for all *C* cells) and the number of trials given by the variant population size $N_{m}$. We similarly stochastically determine the distribution of wild-type viruses across target cells using a multinomial distribution with the event probability of being in a cell given by 1/C (for all *C* cells) and the number of trials given by the wild-type viral population size $N_{w}$. The mean fitness of a viral variant in the context of cellular coinfection can then be calculated in a manner similar to the one specified in Equation (Equation 4). With $F(k_{i},l_{i})$ as the fitness of a viral genome present in cell *i* with $k_{i}$ variant viral genomes and $l_{i}$ wild-type viral genomes, the mean fitness of a viral variant is obtained by considering the stochastically realized viral content in each cell:
(9)$$\bar{e^{\sigma_{m}}}=\frac{\sum_{i=1}^{C}k_{i}F\left(k_{i},l_{i}\right)}{N_{m}}$$


Similarly, the mean fitness of the wild-type virus is given by:
(10)$$\bar{e^{\sigma_{w}}}=\frac{\sum_{i=1}^{C}l_{i}F\left(k_{i},l_{i}\right)}{N_{w}}$$


We then use Equation (Equation 6) to project the frequency of the viral variant in the next generation. Calling this projected frequency $p^{m}\left(t_{g+1}\right)$, we generate a stochastic realization of this frequency by letting the variant effective population size $N_{m}$ be drawn from a binomial distribution with *N* trials and a probability of success of $p^{m}\left(t_{g+1}\right)$. The realized frequency of the viral variant, $q^{m}\left(t_{g+1}\right)$, in generation $t_{g+1}$ is then given by $N_{m}$/*N*.

### 2.3. Simulated Data

We simulated the models described above to ascertain the effect of cellular coinfection on variant allele frequency changes at various levels of coinfection. We also simulated mock datasets and used them to test the statistical inference methods described in detail below. We simulated one mock dataset using the deterministic within-host evolution model, with observed variant allele frequencies that include measurement noise (noise that is due to an inaccurate measuring process, rather than underlying noise in the viral dynamic process). To implement measurement noise, we let the *observed* variant allele frequency in generation $t_{g}$, $q_{o}^{m}\left(t_{g}\right)$, be drawn from a beta distribution with shape parameter α = $\nu q^{m}\left(t_{g}\right)$ and shape parameter β = $\nu (1-q^{m}\left(t_{g}\right))$:
$$q_{o}^{m}\left(t_{g}\right)∼Beta\left(\nu q^{m}\left(t_{g}\right),\nu \left(1-q^{m}\left(t_{g}\right)\right)\right)$$
where ν quantifies the degree of measurement noise. The parameter ν is constrained to be positive, with higher values corresponding to less measurement noise. We simulated a second mock dataset using the stochastic within-host model, similarly assuming beta-distributed measurement noise.

### 2.4. Empirical H5N1 Data

As an application of the approaches developed here, we used longitudinal allele frequency data from an influenza A subtype H5N1 experimental challenge study in ferrets [23]. We specifically focused on inferring the relative fitness of a single nucleotide variant on the hemagglutinin gene segment (G788A) in the VN1203-HA(4)-CA04 virus. This variant was present in the viral inoculum stock at a frequency of 4.40% and increased in frequency over the course of infection in each of the four ferrets that were challenged with this inoculum. Although G788A allele frequencies were measured in [23] on days 1, 3, and 5 post-inoculation, we excluded the day 5 samples from our analyses. This is because up to (and including) day 3, the viral population in each of the four ferrets exhibited low levels of genetic diversity, with G788A being the only variant present at substantial frequencies. By day 5, additional variants on the hemagglutinin gene segment had emerged, with some reaching high frequencies. Because there is genetic linkage between these later variants and G788A, the G788A frequency changes between days 3 and 5 are likely due in part to selection acting on these later variants. Because our model does not reconstruct viral haplotypes or consider epistatic interactions between loci, we thus decided to exclude day 5 from our analysis to be able to focus more specifically on estimating the fitness of G788A in the context of cellular coinfection.

### 2.5. Statistical Inference

The deterministic within-host model contains four parameters: the relative fitness of the variant virus ($e^{\sigma_{m}}$) over the wild-type virus, the mean cellular multiplicity of infection (*M*), the initial frequency of the variant virus in a host ($q^{m}\left(t_{0}\right)$), and the magnitude of measurement noise (ν). When interfacing this model with longitudinal allele frequency data, we estimate the first three parameters but do not estimate ν. We do not estimate ν because it can be parameterized from allele frequency measurements from replicate samples. To estimate $e^{\sigma_{m}}$, *M*, and $q^{m}\left(t_{0}\right)$, we rely on Markov Chain Monte Carlo (MCMC) approaches.

The stochastic within-host evolution model contains the same four parameters as the deterministic model, and one additional parameter: the effective viral population size *N*. (The number of target cells *C* is not an additional parameter because it is given by the product of the effective viral population size *N* and the mean cellular MOI *M*.) We use particle MCMC (pMCMC) [24] to infer $e^{\sigma_{m}}$, *M*, and $q^{m}\left(t_{0}\right)$, and set ν and *N* as given. Particle MCMC is a Bayesian inference approach that combines particle filtering with MCMC to estimate parameters of stochastic state-space models and to reconstruct unobserved state variables. This statistical inference method is increasingly used in the infectious disease modeling community [25,26] but as of yet has not been applied to within-host viral models.

For both the deterministic and stochastic within-host models, let $P(q_{o}^{m}\left(t_{g}\right))$ be the probability of observing a variant allele frequency of $q_{o}^{m}$ in generation $t_{g}$. This probability is given by the beta probability density function, with shape parameters $\nu q_{sim}^{m}\left(t_{g}\right)$ and $\nu (1-q_{sim}^{m}\left(t_{g}\right))$, evaluated at $q_{o}^{m}\left(t_{g}\right)$, where $q_{sim}^{m}\left(t_{g}\right)$ is the model-simulated allele frequency in generation $t_{g}$. This simulated variant allele frequency depends on parameters $e^{\sigma_{m}}$, *M*, and $q^{m}\left(t_{0}\right)$, and for the stochastic model also *N*. For the deterministic model, the likelihood of the model is then given by:
(11)$$\prod_{g}P\left(q_{o}^{m}\left(t_{g}\right)\right)$$
where *g* indexes the generation times of all the measured variant allele frequency data points. For the stochastic model, $P(q_{o}^{m}\left(t_{g}\right))$ is used to calculate the particle weights in the pMCMC algorithm.

Statistical inference code was implemented using Python 3.7.4 and Matlab R2020A and is available from https://github.com/koellelab/withinhost_fitnessInference.

## 3. Results

### 3.1. The Extent of Cellular Coinfection Impacts Variant Frequency Dynamics

The within-host models developed above differ from previous models focused on within-host viral evolution by incorporating the possibility of cellular coinfection and its effects on variant frequency dynamics. Simulations of our deterministic model show, as expected, that a beneficial mutation does not increase in frequency as rapidly when cellular coinfection levels are high compared to when they are low (Figure 1A). Our simulations also show that a deleterious mutation does not decrease in frequency as rapidly when cellular coinfection levels are high compared to when they are low (Figure 1B). Both of these effects are a direct consequence of phenotypic hiding that occurs in cells that are infected by more than one viral genome.

Our stochastic within-host evolution model recapitulates the general patterns observed in simulations of the deterministic model, with demographic stochasticity playing a more pronounced role at lower effective viral population sizes, as expected (Figure 1C,D).

### 3.2. Statistical Estimation of Variant Fitness Using the Deterministic Within-Host Model

#### 3.2.1. Statistical Inference with Simulated Data

We first aimed to determine if longitudinal allele frequency data could be used to infer variant fitness in the context of cellular coinfection under the assumption of deterministic within-host evolutionary dynamics. We therefore first generated a mock dataset by forward simulating the deterministic model and adding measurement noise (Figure 2A). Prior to applying the MCMC methods described above to this mock dataset, we assessed the identifiability of the two parameters of greatest biological interest: variant fitness $e^{\sigma_{m}}$ and mean cellular mulitiplicity of infection *M*. We did this by setting the magnitude of measurement noise ν and the initial mutant allele frequency $q^{m}\left(t_{g}=0\right)$ to their true values and plotting the model likelihood over a range of MOIs and over a range of variant fitnesses. Our results indicate that there is a likelihood ‘ridge’ from low MOI-low fitness parameter combinations to high MOI-high fitness parameter combinations (Figure 2B). The presence of this likelihood ridge is expected, given that higher variant fitness in the context of higher MOI compensates for the phenotypic hiding phenomenon that does not occur at lower MOI.

Given this likelihood ridge, it would be difficult to use MCMC to obtain posterior distributions of the model parameters without an informative prior on either variant fitness or MOI. We decided, for the sake of illustration, to adopt a prior on MOI. Specifically, we assumed a lognormal prior on MOI, with a mean of log(2) and a standard deviation of 0.5. We ran the MCMC chain for 20,000 iterations (Appendix A). Posterior distributions for the initial frequency of the variant, MOI, and variant fitness are shown in Figure 2C–E. All true parameters fell within the 95% credible intervals of the estimated parameter values. In Figure 2A, we further plot 10 forward simulations, parameterized with draws from the posterior distributions, alongside the mock data. These results indicate that the deterministic within-host evolution model can be successfully interfaced with longitudinal variant allele frequency data to infer model parameters using MCMC.

#### 3.2.2. Statistical Inference with Experimental H5N1 Challenge Study

We now apply the same MCMC approaches to experimental data from an influenza A subtype H5N1 challenge study performed in ferrets. Figure 3A shows the frequencies of the G788A variant that was present in the inoculum stock at a frequency of 4.40% and increased in all four of the experimentally infected ferrets. For the reasons provided above, we used only days 1 and 3 for estimation of variant fitness. We also used the measured stock frequency of 4.40% as the day 0 data point for all ferrets. While technically the stock frequency and the ferrets’ day 0 data points constitute very different samples, we felt comfortable with this assumption because of the likely very large transmission bottleneck size between the inoculum and index ferrets. Although an estimate of this transmission bottleneck size is not reported on in [23], a study using barcoded virus found that three-quarters of viral barcodes present in the inoculum were transmitted to index (donor) ferrets in experimental challenges that used 10^4^ plaque-forming units (p.f.u.) of virus inoculum [27], which is two orders of magnitude less virus than used in [23]. In the barcoded virus study, some of the barcodes that were transmitted had frequencies as low as 0.5% in the stock, indicating that the transmission bottleneck size was likely hundreds to thousands of virions. Under the assumption of random sampling of virions from the stock, this means that the frequency of G788A on day 0 of the ferrets was likely very close to 4.40%, with measurement noise significantly outweighing any noise stemming from the wide transmission bottleneck. Indeed, calculations involving the binomial distribution (for the transmission bottleneck) and the beta distribution (for measurement noise) indicate that measurement noise dominates transmission bottleneck process noise when the bottleneck size is larger than the measurement noise parameter ν. With a bottleneck size in the hundreds to thousands and the value of ν we use for this dataset (see below), measurement noise is much larger than transmission bottleneck size noise, therefore allowing us to make the assumption that the day 0 allele frequencies of G788A in the ferrets is equal to the stock frequency of G788A.

In fitting our model to these data, we first converted days post inoculation to viral generations by assuming an 8 h influenza virus generation time based on [28]. Replicate samples for this experiment were not available, so we set the degree of measurement noise ν to 100, but consider the sensitivity of our results to this value (see below). We used an informative prior on the mean cellular MOI, specifically a lognormal prior with a mean of log(4) and a standard deviation of 0.4. We used this prior based on studies that indicate that 3–4 virions are generally required to yield progeny virus from an infected cell [11]. However, we note that a wide range of estimates exist in the literature on the extent of viral complementation required for successful influenza virus progeny production, with findings indicating that this depends on the host cell type and on the viral strain considered [10,12]. We ran the MCMC chain for 20,000 iterations (Appendix A). Posterior distributions for mean cellular MOI and variant fitness are shown in Figure 3B,C, respectively. The joint density plot of MOI and variant fitness (Figure 3D) indicates that there is a positive correlation between these two parameters, consistent with our findings on simulated data (Figure 2B). Posterior distributions for the initial frequencies of the variant in each ferret are shown in Appendix A.

The results shown in Figure 3B indicate that cellular MOI is relatively high, although the informative prior used played a large role in shaping this parameter’s posterior distribution. Our estimate of variant fitness (relative to wild-type fitness) lies between 2.11 and 7.91, with a median value of 3.15. This stands in stark contrast to a previous fitness estimate for this variant of approximately $e^{0.35}=1.42$ [9]. However, this previous estimate was based on a model that did not consider cellular coinfection. With high levels of coinfection thought to occur in within-host influenza virus infections [11] and our inference of relatively high cellular MOI (Figure 3B), higher fitness was inferred for G788A to be able to account for its observed rapid rise in the context of phenotypic hiding. Indeed, the joint density plot shown in Figure 3D indicates that if we had constrained MOIs to be lower (closer in line with the estimates from [10]), our variant fitness estimates would have been considerably closer to those previously inferred for G788A.

Our inferred fitness estimate of ∼2–8 for G788A may initially seem unreasonably large. However, several studies that have estimated variant fitness using in vitro experiments have arrived at estimates of similar magnitude. For example, a recent in vitro study of dengue virus evolution performed at low MOI found that, of the beneficial mutations that were identified, some had relative fitness effects exceeding 2 [29]. An in vitro study focused on HIV similarly found that beneficial mutations could have pronounced effects on viral fitness, with the largest estimated relative fitness of a single mutation being 6.6 [30]. These studies show that the fitness effects of viral mutations can be quite high, particularly when under strong selection pressure. While our relative fitness estimate of ∼2–8 for G788A falls in the range of other estimates present in the viral literature, there are also studies that have inferred lower fitness values for beneficial mutations. For example, the highest relative fitness value estimated for an influenza B mutation that conferred resistance to a neuraminidase inhibitor was 1.8 [31].

The results presented in Figure 3 assume measurement noise ν of 100 and a viral generation time of 8 h. To ascertain the effects of these assumptions on our results, we first re-estimated MOI, variant fitness, and initial variant frequencies under the assumption of both higher (ν = 25) and lower (ν = 400) levels of measurement noise (Appendix A). With higher levels of measurement noise, 95% credible interval ranges for MOI and variant fitness were both wider than when measurement noise was set to ν = 100. In contrast, with lower measurement noise, 95% credible interval ranges for MOI and variant fitness were both considerably more narrow than when measurement noise was set to ν = 100, with variant fitness estimates falling in the range of 2.25–3.6. At both higher and lower levels of measurement noise, median estimates for MOI and variant fitness were not considerably impacted. We also considered the sensitivity of our results to the viral generation time assumed (Appendix A). With a shorter generation time of 6 h, the posterior distribution for MOI remained similar to one inferred using a viral generation time of 8 h. However, variant fitness estimates were lower, with the 95% credible interval range of 1.66–3.47 and a median value of 2.36. With a longer generation time of 12 h, the posterior distribution for MOI again remained similar to one inferred using a viral generation time of 8 h. Variant fitness estimates using a 12 h viral generation time were considerably higher, however, with the 95% credible interval range of 2.88–9.31 and a median value of 4.58. These results underscore the importance of accurately parameterizing the viral generation time when performing variant fitness estimation.

In Figure 4, we show 10 forward simulations of the deterministic model, parameterized using draws from the posterior distributions. These indicate that the model, simulated using parameter estimates inferred from MCMC, reproduces observed G788A allele frequency patterns on days 0, 1, and 3 (the days included in the statistical analyses). The model, however, significantly over-predicts G788A frequencies on day 5 in ferret 15 and ferret 21 (Figure 4B,D). It is interesting to note that in both ferrets 15 and 21, one additional variant (G738A) rose to high frequencies between days 3 and 5. Previous work has inferred a large relative fitness value for this variant ($e^{0.9}=2.5$) as well as (slightly negative) epistatic interactions between it and G788A [9]. Haplotype reconstruction indicates that the ‘A’ allele at site 738 arose in the genetic background of the ‘G’ allele at site 788 [9,23]. With the ‘A’ allele at site 738 conferring a large fitness advantage, and its genetic linkage to the ‘G’ allele at site 788, we would anticipate that this mutation would slow or even reverse the rise of variant G788A between days 3 and 5 in these ferrets due to this process of clonal interference. Indeed, our model projections significantly overestimate the frequency of G788A on day 5 in both of these ferrets, indicating that selection efficiently acted on G738A, impeding the projected increase in the frequency of G788A between days 3 and 5. It is also interesting to examine the dynamics of additional variants in ferrets 13 and 17, where the model predicts G788A frequencies relatively well on day 5, although this data point was not used during model fitting. Ferret 13 had one other variant arising between day 3 and day 5 (variant G496T). A previous study using these data inferred a large relative fitness value for this variant ($e^{0.7}=2.0$) [9]. Our model simulations, however, projected the allele frequency of G788A on day 5 well in the absence of considering this variant. As such, we would predict that this G496T variant had lower relative fitness than previously estimated. Ferret 17 also had one other variant rising to high frequencies between day 3 and day 5 (variant C736A). It is unclear whether previous work inferred this mutation to be strongly beneficial or strongly deleterious, since A736C (rather than C736A) was the mutation identified as being under positive selection. Regardless, our model slightly over-projects the frequency of G788A on day 5, such that we expect C736A to have contributed to some extent to allele frequency changes of G788A through linkage effects.

In Figure 4, we further plot model simulations that assume no cellular coinfection. Specifically, we simulate Equation (Equation 1) where the dynamics are driven by the variant’s individual-level fitness $e^{\sigma_{m}}$ rather than by $\bar{e^{\sigma_{m}}}$. The frequency of G788A rises considerably faster in these simulations compared to those that incorporate cellular coinfection. This indicates that the speed of within-host viral adaptation is severely reduced by cellular coinfection.

### 3.3. Statistical Estimation of Variant Fitness Using the Stochastic Within-Host Evolution Model

#### 3.3.1. Statistical Inference with Simulated Data

The within-host evolutionary dynamics of viral pathogens may not be appropriately described by a deterministic model, even though viral population sizes within infected individuals over the course of infection are often times very large. Indeed, recent studies have highlighted the role that stochastic processes play in within-host viral dynamics [21,22]. We therefore next aimed to determine if longitudinal allele frequency data could be used to infer variant fitness in the context of cellular coinfection under a model of within-host evolution that incorporated stochastic effects. As described above, we incorporated stochastic effects by implementing the within-host model with a small effective viral population size, specified by the parameter *N*. As such, we are modeling demographic stochasticity, with genetic drift being the driver of random changes in variant allele frequencies. We generated a mock within-host dataset by forward-simulating the stochastic model and adding measurement noise (Figure 5A). This dataset was generated under the same parameterization as the deterministic dataset shown in Figure 2A, with stochastic effects included by setting the viral population size *N* to 100.

Using the same prior on MOI as with the deterministic analysis and under the assumption that ν and *N* are known, we ran the MCMC chain for 50,000 iterations (Appendix A). Posterior distributions for the initial frequency of the variant, mean cellular MOI, and variant fitness are shown in Figure 5B–D. All true parameters fell within the 95% credible intervals of the estimated parameter values. In Figure 5A, we plot 10 reconstructed variant allele frequency trajectories. These results indicate that the stochastic within-host evolution model can be successfully interfaced with longitudinal variant allele frequency data to infer model parameters and underlying variant frequency dynamics using pMCMC.

#### 3.3.2. Statistical Inference with Experimental H5N1 Challenge Study Data

We now apply the same pMCMC approaches to the H5N1 experimental data analyzed already using the deterministic model. We again used only days 0, 1, and 3 for estimation of variant fitness, assumed a viral generation time of 8 h, and set the degree of measurement noise ν to 100. We used the same informative prior on the mean cellular multiplicity of infection *M*. We set the effective viral population size to N=100 in our analyses, but consider a scenario of even higher stochasticity below. We ran the pMCMC chain for 50,000 iterations (Appendix A). Posterior distributions for mean cellular MOI and variant fitness are shown in Figure 6A,B, respectively. The joint density plot of MOI and variant fitness (Appendix A) again indicates that there is a positive correlation between these parameters, consistent with the results from our analysis using the deterministic model. Posterior distributions for the initial frequencies of the variant in each ferret are shown in Appendix A. Our results are consistent with the findings from our deterministic analysis: we estimate that the fitness of the G788A variant is considerably (2–10.5 times) higher than that of the wild-type virus. These results are robust to higher levels of stochasticity; we show, for example, the posterior distributions for *M* and variant fitness under the assumption that the viral effective population size *N* is 40 (Appendix A).

## 4. Discussion

Here, we developed mathematical within-host models that can take into consideration cellular coinfection when projecting changes in viral allele frequencies over the course of an infection. We further described and demonstrated how these evolutionary models can be statistically interfaced with viral sequence data to jointly estimate variant fitness relative to the wild-type allele along with the mean cellular multiplicity of infection. Our results indicate that ignoring the possibility of cellular coinfection can result in significant underestimation of a variant’s selective advantage. This is important because a variant with a much higher selective advantage, once established monomorphically within a host, is expected to have a more precipitous impact on within-host viral dynamics than a variant with a smaller selective advantage. We might, for example, expect a variant with a higher selective advantage to result in higher peak viral loads and potentially longer durations of infection. This would impact both symptom development as well as onward transmission potential.

Our models, like all models, make some simplifying assumptions. First, we assume low viral diversity, with diversity comprising just one locus and two alleles (a wild-type and a variant allele). We chose to model evolution at a single locus to highlight the important contribution that cellular coinfection may play in the within-host evolution of viral pathogens. Our application to the G788A mutation in the H5N1 experimental challenge study in ferrets satisfied this assumption between days 0 through 3. Because other sites became polymorphic in each of the four studied ferrets by day 5, we excluded this time point from our statistical analyses. To consider the effect of cellular coinfection within a system with higher levels of genetic diversity, and the possibility of new variants arising over the course of infection, the models developed here should be extended using approaches developed already in [9]. These approaches include inference of viral haplotypes and the incorporation of de novo mutations into the presented model structures. With these additions, full genetic linkage between loci can be considered, and epistatic interactions between loci can also be inferred. Our models, as presented here, however, could still be applied to higher diversity viral systems if recombination occurred freely between loci, as may be the case between influenza gene segments or some viruses with high recombination rates.

A second assumption present in the current formulation of our models is that viral fitness is additive: if a coinfected cell harbors both variant and wild-type viral genomes, then the fitness of each viral genome is not only assumed to be equal, but also equal to the arithmetic mean fitness of the involved genomes. This may be a good assumption if the focal mutation impacts, for example, polymerase activity, with the viral polymerase protein being used for the replication of all viral genomes. However, it may also be the case that a mutation has a disproportionate effect on intracellular viral fitness. Future work should therefore examine the impact of a mutation’s ‘dominance’ [32] on in vivo viral evolution.

A third assumption is one that is somewhat less transparent in the structure of our models, namely that we assume that there is no intracellular viral competition for host cell machinery. This assumption is reflected in the calculation of a variant allele’s mean fitness ($\bar{e^{\sigma_{m}}}$). A single viral genome’s fitness in a cell depends on the genotypes of the other genomes present in the cell, but not on the cellular multiplicity of infection directly. If a variant genome is in a cell alone or with a large number of other variant genomes, for example, its fitness will be the same. However, if host cell machinery is limiting, one would expect the per genome fitness—which can be interpreted here as *per capita* viral yield or reproductive success—to be lower in highly coinfected cells. Indeed, empirical studies with influenza virus indicates that there is a saturating relationship between viral input and viral output from a cell [33]: at low cellular MOI, doubling the viral input yields a doubling of viral output, such that viral competition is not readily apparent; at high MOI, however, doubling the viral input does not appreciably change the overall viral output, indicative of limiting host cell machinery. Future work should therefore also examine the impact of intracellular viral competition on within-host viral evolution and extend models such as the ones we presented here to account for intracellular viral competition.

Finally, our model assumes that the mean cellular multiplicity of infection (MOI) is fixed across viral generations and that virion entry into cells is governed by a Poisson process. In terms of the former assumption, it is conceivable that MOI might change over the course of an infection. For example, at the beginning of a viral infection, MOI may be low because a very small viral population is initiating infection in a large environment of host cells. As viruses replicate within their host, viral population sizes increase and the number of target cells decreases. This may result in more individual-level selection at the beginning of the infection (due to low MOI), followed by a greater degree of phenotypic hiding later on in the infection (due to higher MOI). To accommodate these changes in MOI, the structures of the within-host models presented here would not need to be significantly altered; MOI could simply be made into a time-varying parameter. For simplicity, we here instead decided to assume that MOI is fixed over the course of infection, in part because of the lack of empirical data to inform MOI at multiple time points over the course of an infection. A further argument against incorporating dynamic changes in MOI is that spatially structured within-host viral dynamics, such as those characterized for influenza [34], may result in cellular MOIs that are more uniform over time than expected from a spatially unstructured setting. In terms of the latter assumption (Poisson-distributed virions), there are several reasons why this assumption may not be met. Virions could aggregate, such that virion entry into cells is not an independent process. Cells could also be heterogeneous with respect to their susceptibility to infection, for example due to their cell cycle state or due to antiviral states triggered by interferon. Both of these factors would result in virions being overdispersed across cells, rather than Poisson-distributed. While considering different assumptions of how virions are distributed across cells is beyond the scope of this study, future work should address the effect of viral overdispersion on variant fitness estimation.

Despite these limiting assumptions, a general takeaway from the evolutionary models presented here is that cellular coinfection will slow down the rate of viral adaptation within hosts when adaptation occurs through selection acting on single point mutations (or insertions/deletions) as we considered here. (A caveat here is that cellular coinfection could accelerate viral adaptation if it heavily relies on genetic exchange, i.e., recombination or reassortment.) Slower rates of viral adaptation is good news from the perspective of the host population, as this will also slow down viral adaptation at the population-level. This finding has clear implications for emerging zoonotic viruses that are adapting to a new host population. Analogously, cellular coinfection will result in less effective purging of deleterious mutations. By making natural selection a weaker evolutionary force, cellular coinfection may thus be one reason why stochastic processes appear to dominate within-host viral dynamics and why selection does not seem to act efficiently over the course of an acute infection for viruses such as seasonal influenza [21,35]. There are other factors, however, that may also limit the ability for positive selection to act efficiently within hosts. For example, the temporal asynchrony between the timing of the immune response and when virus diversification occurs may explain why antigenic immune escape variants do not readily arise in individuals with some pre-existing immunity [36]. A second takeaway is that variants whose fitness levels (relative to wild-type) have been quantified using models that do not include cellular coinfection may have significantly underestimated variant fitness. Underestimation of variant fitness may underestimate the effect of a mutation on viral replication dynamics once those dynamics involve only the variant virus. Our results—that the fitness effect of certain mutations can be large—speak to the adaptive potential of these viruses to new or changing host populations, even if adaptation may occur more slowly than might be expected.

## Figures and Tables

**Figure 1 viruses-13-01216-f001:**
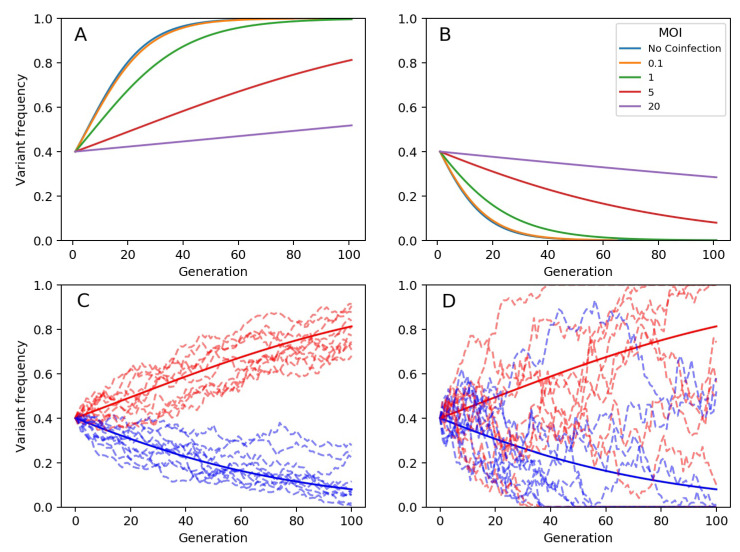
Model simulations showing changes in variant frequencies over viral generations. (**A**) Frequency changes of a beneficial mutation under the deterministic within-host model, parameterized at different mean cellular multiplicities of infection *M*. For all simulations shown, the variant’s fitness is $e^{\sigma_{m}}=1.1$ and its initial frequency is $q^{m}\left(t_{0}\right)=0.4$. (**B**) Frequency changes of a deleterious mutation under the deterministic within-host model, parameterized at different mean cellular multiplicities of infection *M*. For all simulations shown, the variant’s fitness is $e^{\sigma_{m}}=0.9$ and its initial frequency is $q^{m}\left(t_{0}\right)=0.4$. In (**A**,**B**), we consider MOI values of 0.1, 1, 5, and 20. Labeled as ‘No coinfection’, we also plot simulations of the model presented in Equation (Equation 1), which assumes that fitness is an individual-level property of a viral genome. (**C**) Frequency changes of a beneficial mutation (red; $e^{\sigma_{m}}=1.1$) and of a deleterious mutation (blue; $e^{\sigma_{m}}=0.9$) under the stochastic within-host model, parameterized with a mean cellular MOI of 5. Dashed lines show 10 stochastic realizations under each parameterization. Solid lines show simulations of the deterministic model under the same parameterization. Stochastic simulations used an effective viral population size of N=1000. (**D**) Frequency changes of mutations, as in (**C**), only using an effective viral population size of N=100.

**Figure 2 viruses-13-01216-f002:**
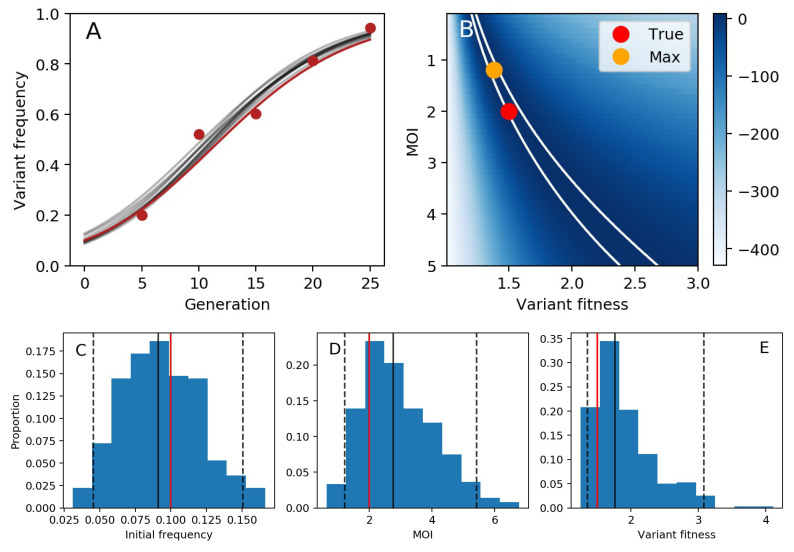
Variant fitness estimation under the assumption of deterministic evolutionary dynamics. (**A**) Mock data (red dots) generated from a forward simulation of the deterministic within-host evolution model with added measurement noise. The underlying deterministic dynamics are shown with a red line. The model is parameterized with variant fitness of $e^{\sigma_{m}}=1.5$, a mean cellular MOI of M=2.0, and an initial frequency of the variant of $q^{m}\left(t_{0}\right)=0.10$. Measurement (observation) noise is set to ν=100. Grey lines show 10 model simulations, with parameters drawn from the MCMC posterior distributions. (**B**) Log-likelihood landscape, showing the log-likelihood of the model over a broad range of MOI and variant fitness values. When calculating these likelihoods, the initial frequency of the variant and the measurement noise were fixed at their true values. The red dot shows the true set of parameters used to simulate the mock data. The yellow dot shows the parameter combination yielding the highest log-likelihood. White boundary lines show the 95% confidence interval of parameter estimates. (**C**) Posterior distribution for the initial frequency of the variant. (**D**) Posterior distribution for the mean cellular multiplicity of infection *M*. (**E**) Posterior distribution for variant fitness. In (**C**–**E**), black solid lines show the median values of the posterior density, black dashed lines show the 95% credible intervals, and red solid lines show the true values.

**Figure 3 viruses-13-01216-f003:**
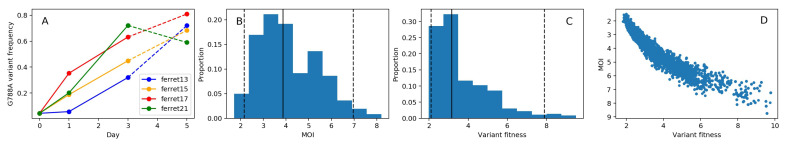
Fitness estimation for variant G788A, assuming deterministic within-host dynamics. (**A**) Measured G788A allele frequencies over the course of infection for 4 experimentally infected ferrets. Days 0 (stock frequency), 1, and 3 are used in the estimation of variant fitness. (**B**) Posterior distribution for the mean cellular multiplicity of infection. (**C**) Posterior distribution for variant fitness. In (**B**,**C**), black solid lines show the median values of the posterior densities and black dashed lines show the 95% credible intervals. (**D**) Joint density plot for MOI and variant fitness.

**Figure 4 viruses-13-01216-f004:**
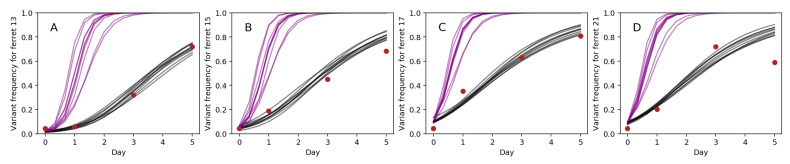
Deterministic model simulations (grey lines) and observed data points (red dots) are shown for (**A**) ferret 13, (**B**) ferret 15, (**C**) ferret 17, and (**D**) ferret 21. Only days 0, 1, and 3 were used in model fitting. Parameters for the model simulations were drawn from the posterior distributions of the parameters. Purple lines show model simulations under the same parameterizations of variant fitness and initial variant frequencies as the grey lines, but simulated in the absence of cellular coinfection. These no-coinfection projections were simulated using Equation (Equation 1).

**Figure 5 viruses-13-01216-f005:**
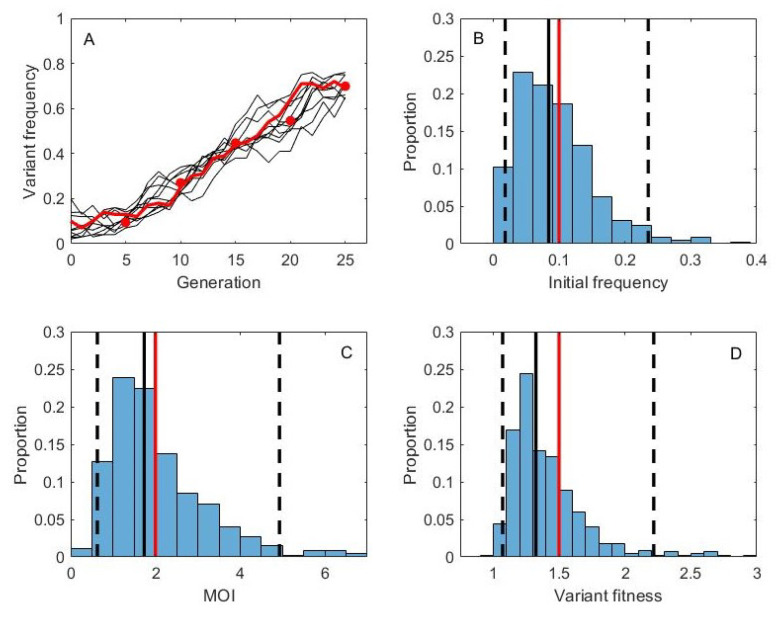
Variant fitness estimation under the assumption of stochastic evolutionary dynamics. (**A**) Mock data (red dots) generated from a forward simulation of the stochastic within-host evolution model with added measurement noise. The underlying stochastic dynamics are shown with a red line. The model is parameterized with variant fitness of $e^{\sigma_{m}}=1.5$, a mean cellular MOI of M=2.0, an initial frequency of the variant of $q^{m}\left(t_{0}\right)=0.10$. Measurement (observation) noise is set to ν=100. The viral effective population size is set to N=100. Grey lines show the dynamics of 10 reconstructed allele frequency trajectories. These trajectories are unobserved state variables that have been reconstructed using pMCMC. (**B**) Posterior distribution for the initial frequency of the variant. (**C**) Posterior distribution for the mean cellular multiplicity of infection *M*. (**D**) Posterior distribution for variant fitness. In (**B**–**D**), black solid lines show the median values of the posterior densities, black dashed lines show the 95% credible intervals, and red solid lines show the true values. 200 particles were used in the pMCMC.

**Figure 6 viruses-13-01216-f006:**
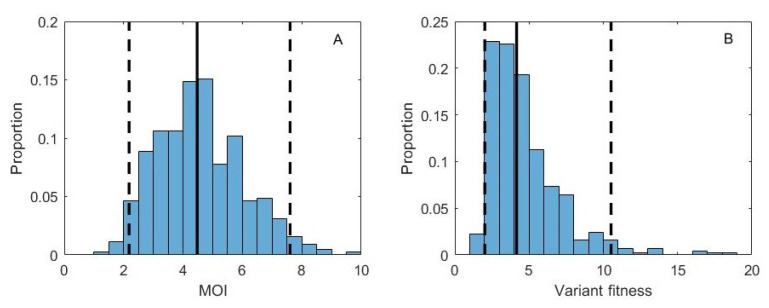
Fitness estimation for variant G788A, assuming stochastic within-host dynamics. (**A**) Posterior distribution for the mean cellular multiplicity of infection *M*. (**B**) Posterior distribution for variant fitness. In (**A**,**B**) the model was parameterized with an effective viral population size of N=100. In (**A**,**B**) black solid lines show the median values of the posterior densities and black dashed lines show the 95% credible intervals.

## Data Availability

Simulated data from the study can be generated using code available at https://github.com/koellelab/withinhost_fitnessInference. Empirical H5N1 data are from [23], with exact data points provided by Dr. Thomas Friedrich upon request.

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
