# Peer review of "Fitness Estimation for Viral Variants in the Context of Cellular Coinfection"

_viruses, 2021, doi:10.3390/v13071216_

Round 1
Reviewer 1 Report
Summary
This is an extremely useful paper that leverages a nuanced understanding of influenza cellular infection dynamics to improve methods assessing the fitness effects of virus mutations from longitudinal with-in host sequencing data. I found the introduction and explanation of the rationale for the method to be particularly well-written. The authors responsibly addressed many of the model limitations, but there were a couple of model assumptions that weren’t well discussed—in particular, some of those used to fit to the experimental H5N1 challenge study. Overall, though, the paper was well constructed and corrects a significant limitation in current methods assessing mutational fitness effects from sequential sequence data. I think that it could be improved with the following recommendations, but I look forward to seeing it published.
Major
- The authors use the 4.4% variant frequency in the stock solution as a measure of day 0 variant frequency within host. However, variant frequency in the inoculum compared to what actually infected the host would have been strongly affected by a transmission bottleneck. What is the intranasal transmission bottleneck and is it wide enough that 4.4% would actually be the variant’s likely frequency amongst the population that established itself? Are the estimates of variant fitness strongly affected by frequency changes between stock solution and day 1 or are these consistent with day 1-3 changes?
- The authors use an experimentally derived measure of the minimum number of particles needed to produce a full infectious virion to set the prior distribution. However, this metric has been shown to vary between influenzas strains (see [10] Brooke et al. 2013). As the estimate of fitness strongly depended on this parameter, the authors should include an explanation of the variation in this parameter and further discuss how it affected fitness predictions.
Minor
- Line 41: The key in these experiments was that virus evolved increased replication rates in transmission relevant tissues (nasal turbinates) while decreasing replication rates in pathogenesis relevant tissues (trachea), not necessarily that they had overall increases in replication rate (they did have overall transmission rate increases).
- Line 103-104 : Specify that this is the within-host effective population size of seasonal influenzas, not global.
- Line 210: Needs a citation for the 8 hour generation time assumption
- Line 126: Specify what you mean by measurement noise when you introduce it
- Line 266-268: This would be called clonal interference
- Line 381-382: Saying that cellular coinfection will slow down the rate of virus adaptation seems a bit of an overreach. While this paper convincingly shows that it should slow the speed of selection on a single allele, cellular coinfection also alters other potentially adaptive processes like reassortment or recombination. Thus, its effect on the overall rate of virus adaptation is not as clear.
- Line 386-388: Recent work (Morris et al. 2020 eLife 10.7554/eLife.62105) also suggests that this inefficiency of natural selection is due to the temporal dynamics of immune activation.
Author Response
Please see attached reply letter document.

Reviewer 2 Report
This manuscript show the development of a mathematical model to estimate viral fitness based on MOI. The model is compared to actual sequence data from a ferret H5N1 infection model, and conclude that co-infection changes host.viral evolutionary dynamics.
The manuscript is well written, with mathematical modelling made reasonably understandable also for general virologists. The strength of the work lies in the comparison with the actual in vivo study in ferrets with longitudinal sequence data. The main limitation however is the lack of one aspect in the discussion:
Main Comment:
L85: The assumption during early model development - that "viral genomes enter cells independantly of other viral genomes" is well known by viral immunologists not to hold through, due to the cell entering an "antiviral state" triggered by interferon. Although it is understandable that model development will have to start up with such assumptions, this aspects must be discussed, e.g. on page 17-18.
L210: The assumption of "an 8 hour influenza virus generation" is in need of a reference.
L369: "MOI change over the course of infection" is in need of a reference(s)
L381: The general takeway should be somewhat modulated based on considerations of the "Antiviral state".
L67 Spell error: ..fitness effect..
Author Response
Please see the attached word document.
